# Pharmacokinetic/Pharmacodynamic Evaluation of Aztreonam/Amoxicillin/Clavulanate Combination against New Delhi Metallo-β-Lactamase and Serine-β-Lactamase Co-Producing *Escherichia coli* and *Klebsiella pneumoniae*

**DOI:** 10.3390/pharmaceutics15010251

**Published:** 2023-01-11

**Authors:** Jiayuan Zhang, Mengyuan Wu, Shuo Diao, Shixing Zhu, Chu Song, Jiali Yue, Frederico S. Martins, Peijuan Zhu, Zhihua Lv, Yuanqi Zhu, Mingming Yu, Sherwin K. B. Sy

**Affiliations:** 1School of Medicine and Pharmacy, Ocean University of China, 5 Yushan Road, Qingdao 266003, China; 2Faculty of Pharmaceutical Sciences of Ribeirão Preto, University of São Paulo, Ribeirão Preto 14040-900, São Paulo, Brazil; 3Department of Pharmacology, University of Pennsylvania, Philadelphia, PA 19104, USA; 4Laboratory for Marine Drugs and Bioproducts of Qingdao National Laboratory for Marine Science and Technology, Qingdao 266003, China; 5Department of Laboratory Medicine, The Affiliated Hospital of Qingdao University, Qingdao 266003, China; 6Department of Statistics, State University of Maringá, Maringá 87020-900, Paraná, Brazil

**Keywords:** *Klebsiella pneumoniae*, *Escherichia coli*, NDM, MPC, pharmacodynamics

## Abstract

This study aimed to examine specific niches and usage for the aztreonam/amoxicillin/clavulanate combination and to use population pharmacokinetic simulations of clinical dosing regimens to predict the impact of this combination on restricting mutant selection. The in vitro susceptibility of 19 New-Delhi metallo-β-lactamase (NDM)-producing clinical isolates to amoxicillin/clavulanate and aztreonam alone and in co-administration was determined based on the minimum inhibitory concentration (MIC) and mutant prevention concentration (MPC). The fractions of a 24-h duration that the free drug concentration was within the mutant selection window (*f*T_MSW_) and above the MPC (*f*T_>MPC_) in both plasma and epithelial lining fluid were determined from simulations of 10,000 subject profiles based on regimens by renal function categories. This combination reduced the MIC of aztreonam and amoxicillin/clavulanate to values below their clinical breakpoint in 7/9 *K. pneumoniae* and 8/9 *E. coli*, depending on the β-lactamase genes detected in the isolate. In the majority of the tested isolates, the combination resulted in *f*T_>MPC_ > 90% and *f*T_MSW_ < 10% for both aztreonam and amoxicillin/clavulanate. Clinical dosing regimens of aztreonam and amoxicillin/clavulanate were sufficient to provide mutant restriction coverage for MPC and MIC ≤ 4 mg/L. This combination has limited coverage against NDM- and extended-spectrum β-lactamase co-producing *E. coli* and *K. pneumoniae* and is not effective against isolates carrying plasmid-mediated AmpC and KPC-2. This study offers a potential scope and limitations as to where the aztreonam/amoxicillin/clavulanate combination may succeed or fail.

## 1. Introduction

*Klebsiella pneumoniae* and *Escherichia coli* are common multidrug-resistant (MDR) Gram-negative bacilli associated with hospital-acquired and ventilator-associated pneumonia (HAP/VAP) in critically ill patients [1,2]. Infections due to MDR bacteria are responsible for at least 700,000 deaths annually [3]. Even though polymyxins are emerging as the antibiotics of last resort to treat MDR infections including HAP/VAP, this class of drug is associated with toxicity and poor therapeutic outcomes, including a rapid rise of heteroresistance in monotherapy [4]. The combination of polymyxins with other antibiotics in the treatment of MDR infections has exhibited in vitro synergies [5,6,7,8], but there is no convincing clinical evidence that a combination regimen of polymyxins is superior to monotherapy [9].

Several new β-lactam/β-lactamase inhibitor (BL/BLi) combinations were approved in the last decade to treat Enterobacterales harboring class A, C and D β-lactamases. Many of these recently approved BL/BLi combinations such as ceftazidime/avibactam, meropenem/vaborbactam and imipenem/cilastatin/relebactam are not active against the class B metallo-β-lactamases (MBL) [10]. Enterobacterales harboring MBL are resistant to ceftazidime/avibactam [11]. Aztreonam/avibactam is currently being investigated as a treatment option against MBL-producing Gram-negative bacteria [12]. Even though aztreonam is impervious to MBL, the concerted expressions of extended-spectrum β-lactamases (ESBL) in MDR bacteria can hydrolyze aztreonam [13]. Aztreonam combined with old and new β-lactamase inhibitors is being evaluated against MBL and ESBL co-producing Gram-negative clinical isolates as a possible option for the treatment of complicated infections [14].

Plasmid-encoded or chromosomally-encoded ESBL genes are transmissible horizontally to other Enterobacterales, allowing ESBL-producing Gram-negative bacilli to propagate ESBL clones between patients [15,16]. Clonally-related isolates are often detected among patients residing in the same healthcare facilities and geographical regions [17]. During an outbreak situation, Gram-negative bacteria acquired ESBL genes through clonal dissemination [18]. Even during endemic periods, clonally-unrelated strains can harbor identical ESBL-carrying plasmids through horizontal gene transfer between the same Enterobacterales species, as well as between *E. coli* and *K. pneumoniae* [19]. Most of the plasmid-mediated ESBLs are derived from TEM, SHV and CTX-M β-lactamase genes and confer transferable resistance to cephalosporins. Regional differences in antibiotic-resistant pathogens pose further challenges for clinicians.

The β-lactam ring of clavulanic acid binds irreversibly to the bacterial β-lactamase, preventing it from inactivating β-lactam antibiotics. Clavulanic acid interacts with the serine β-lactamase involving a nucleophilic attack of the active-site serine Oγ with the carbonyl carbon of the β-lactam ring, resulting in the formation of an acyl enzyme. Even though the acyl enzyme complex formed between the serine β-lactamase and clavulanic acid is quite stable, the TEM-1 enzyme was shown to recover approximately 50% of its activity within a day [20]. Clavulanic acid is a more potent inhibitor than sulbactam of both conventional-spectrum and extended-spectrum TEM, SHV and OXA enzymes, except OXA-2 [21]. However, clavulanic acid does not inhibit OXA-24, KPC-2, P99 and AmpC β-lactamases as well as NDM [20,22]. Given that aztreonam is not hydrolyzed by NDM, while clavulanate inhibits many of the serine β-lactamases, the combination of aztreonam and amoxicillin/clavulanate can overcome many NDM producers that co-express serine β-lactamases. Recent studies have shown that the combination of aztreonam and amoxicillin/clavulanic acid had a significant synergistic effect against infections due to carbapenemase- and ESBL-producing Enterobacterales, as well as those harboring NDM [23].

The objectives of this study were to examine specific niches and usage for the aztreonam/amoxicillin/clavulanate combination and to evaluate the effect of co-administration of aztreonam and amoxicillin/clavulanate on the pharmacodynamic (PD) indices associated with the selection of resistant mutants against MDR *K. pneumoniae* and *E. coli* strains harboring NDM. 

## 2. Methods

### 2.1. Antimicrobial Agents

Analytical-grade amoxicillin, clavulanic acid and aztreonam were purchased from the Shanghai Macklin Biochemical Co., Ltd. (Macklin Biochemical Co. Ltd., Shanghai, China). The stock solutions of amoxicillin (10.24 mg/mL), clavulanic acid (10.24 mg/mL) and aztreonam (10.24 mg/mL) were prepared according to CLSI guidelines [24].

### 2.2. Bacterial Isolates and Susceptibility Testing

All clinical isolates of *K. pneumoniae* and *E. coli* were provided by the affiliated hospital of Qingdao University. Next-generation sequencing was used to determine the types of β-lactamase gene for each strain, as previously described [11]. Eighteen clinical isolates that carry *bla*_NDM_ were selected for further testing. *E. coli* ATCC 25922 and ATCC 35218, *K. pneumoniae* ATCC 700603 and *C. braakii* HDC438 were used as quality control strains.

A checkerboard method using broth microdilution was used to determine the susceptibility profile of each clinical isolate to aztreonam alone, amoxicillin/clavulanate, and aztreonam/amoxicillin/clavulanate. MIC determination was conducted in triplicate and modal MIC was used to characterize the susceptibility of each isolate to the antibiotics.

Four *K. pneumoniae* and three *E. coli* strains were selected to determine the antimicrobial’s mutant prevention concentration (MPC) against each strain. A series of Mueller–Hinton agar plates containing an antimicrobial agent or combination at 1×, 2×, 4×, 8×, 16× and 32 × MIC were prepared. A high-density inoculum of ≥ 10^10^ cfu/mL was cultured and confirmed by absorbance at 660 nm OD (Milton Roy Spectronic 21D spectrophotometer) to ensure the emergence of more resistant bacteria. The prepared Mueller–Hinton agar was inoculated with a high inoculum density, evenly coated, and cultured at 37 °C for 48 h. The Mueller–Hinton agar was observed and recorded every 24 h. The lowest antimicrobial concentration that completely prevented bacterial growth was determined as the MPC.

### 2.3. Time–Kill Kinetic Assay

In order to study the effects of amoxicillin/clavulanate and aztreonam alone or in combination on the in vitro time-course of bacterial growth and kill dynamics, one strain of each bacteria type was selected to run the time–kill kinetic experiment at MIC and two-fold MIC. There were four treatment groups for each strain: control, amoxicillin/clavulanate, aztreonam, and aztreonam plus amoxicillin/clavulanate. *K. pneumoniae* LW5 and *E. coli* 37 were cultured in Mueller–Hinton broth at 37 °C for an hour to ensure that logarithmic growth was achieved. The initial inoculum size was approximately 5 × 10^5^ cfu/mL. There was no drug added to the control group. For *K. pneumoniae* LW5, the drug concentrations at MIC were 64/32 mg/L amoxicillin/clavulanate, 32 mg/L aztreonam, 1/0.5 mg/L amoxicillin/clavulanate plus 0.25 mg/L aztreonam. For the *E. coli* isolate 37, the MICs were 128/64 mg/L amoxicillin/clavulanate, 32 mg/L aztreonam, 1/0.5 mg/L amoxicillin/clavulanate plus 0.25 mg/L aztreonam. Twice the previous drug concentrations were added in the two-fold MIC treatment arm. The culture flask was incubated at 37 °C and constant shaking at 120 rpm. At 0, 2, 4, 6 and 8 h post-drug administration, 20 µL of the liquid was sampled and diluted in 10-fold increments in fresh sterile saline at room temperature and spread 100 μL volumes from each dilution in 10 spots on agar plates. Colonies were counted following incubation at 37 °C for 18 to 24 h. All time–kill experiments were performed in triplicate on separate occasions. The average concentration of bacteria at each time point was calculated and recorded.

### 2.4. Population Pharmacokinetic Simulations

The dosing regimens of aztreonam and amoxicillin/clavulanate used in the simulation are listed in Table 1. The aztreonam high dose did not exceed 6 g daily doses, whereas the amoxicillin/clavulanate dosing regimens were based on that recommended in the package insert. A prolonged 3 h infusion was used to achieve higher *f*T_>MIC_. The same dosing interval was used for both aztreonam and amoxicillin/clavulanate, assuming that these antimicrobials are administered simultaneously.

A detailed description of the population pharmacokinetic models and setup for simulations are provided in the Appendix A. The demographics for the virtual patient population consisted of males and females in equal proportion with various degrees of renal impairment. Creatinine clearance (CrCL) was simulated using a uniform distribution ranging from 10 to 150 mL/min and normalized to 1.73 m^2^ body surface area. The lower limit of CrCL was set at 10 mL/min to exclude patients who undergo hemodialysis, which can change the clearance of drugs. The steady-state concentration–time profiles of aztreonam, amoxicillin and clavulanate were simulated from the population pharmacokinetic models described below. 

The model for aztreonam in patients with renal impairment and complicated intra-abdominal infection is a two-compartment model [25,26] wherein its clearance (CL in L/h) is dependent on CrCL such that $CL=4.73*{\left(CrCL/100\right)}^{0.43}$ with 24.1% coefficient of variation (CV%). Central volume (V_C_ in L) is dependent on body weight (WT) such that $V_{C}=7.43*{\left(WT/70\right)}^{1.99}$ with 50.9% CV. Peripheral volume (V_P_) is 6.44 L (27.7% CV) and intercompartmental clearance (Q) is 9.26 L/h. The plasma protein binding of aztreonam is 56%. 

The population pharmacokinetic models of amoxicillin and clavulanate in critically ill patients were previously described [27]. Amoxicillin concentration–time profiles were simulated using a two-compartment IV model: $CL=10.3*\left(\frac{CrCL}{102}\right)$, CV = 39.9%; $V_{C}=13.5\;$, CV = 38.7%; $V_{P}=14.1$; and Q=15.7. Amoxicillin plasma protein binding is 17%. The model of clavulanate is also a two-compartment IV model such that:$\;CL=6.8*\left(\frac{CrCL}{102}\right)$, CV = 57.8%; $V_{C}=7.6\;$, CV = 34.7%; $V_{P}=11.6$; and Q=10.4. Clavulanate is approximately 25% protein bound.

The drug concentration in the lung was approximated based on epithelial lining fluid (ELF) penetration rates. These rates were 55% and 30% for aztreonam and amoxicillin/clavulanate, respectively [28,29]. No mucin binding was assumed for drug exposures in the ELF.

### 2.5. Pharmacodynamic Metrics to Quantify Suppression of Emergent Mutant

The restriction of the resistant mutant selection was measured by two PD parameters, *f*T_MSW_ and *f*T_>MPC_, for the fraction of time over the 24 hours wherein the free drug concentration was within the MSW and the fraction of time over the same period wherein the free drug concentration exceeded the MPC, respectively. Limiting the selection of resistant mutants can be achieved by maintaining drug concentrations above MPC during therapy and decreasing the time that the bacteria spent within MSW [17,30,31]. 

Then, 10,000 concentration–time profiles at steady-state over a 24 h period were simulated for each drug to compute these two PD parameters. Both *f*T_>MIC_ and *f*T_>MPC_ were determined for each profile. *f*T_MSW_ was subsequently computed as the difference between *f*T_>MIC_ and *f*T_>MPC_. If *f*T_>MPC_ is 0, *f*T_MSW_ is not computed. For each of the selected isolates, descriptive statistics of the PD parameters were listed.

### 2.6. Target Pharmacodynamic Indices and Probability of Target Attainment

The previous two PD indices associated with the inhibition of drug resistance are dependent on establishing sufficient probability of target attainment (PTA) at the MIC of the combination therapy. For aztreonam, the target PD index associated with 2 log kill in *E. coli* using a single compartment dilutional model was 60% *f*T_>MIC_ [32]. The predefined PD target for amoxicillin/clavulanate was 50% *f*T_>MIC_ [27]. The ability of each dosing regimen to achieve its predefined PD target was determined by PTA at incremental MIC. Dosing simulations for clavulanic acid were not undertaken given that its PD target is not established [27].

### 2.7. Software

All PK simulations and PD analyses were carried out in R (4.1.2) using the RxODE package and user-defined functions, respectively.

## 3. Results

### 3.1. In Vitro Antimicrobial Susceptibility

CLSI breakpoints [24] were used for interpreting amoxicillin/clavulanate MIC results: ≤ 8/4 mg/L (susceptible), 16/8 mg/L (intermediate) and ≥ 32/16 mg/L (resistant); and for aztreonam MIC results: ≤ 4 mg/L (susceptible), 8 mg/L (intermediate) and ≥16 mg/L (resistant). All of the eighteen clinical *K. pneumoniae* and *E. coli* isolates carried NDM along with other extended-spectrum β-lactamase genes, including CTX-M, TEM, OXA, SHV, and plasmid-mediated AmpC (CMY); these isolates exhibited significant drug resistance to either amoxicillin/clavulanate or aztreonam (Table 2). The control *E. coli* ATCC 25922 was susceptible to both aztreonam and amoxicillin/clavulanate, whereas the control *E. coli* ATCC 35218 strain that contained TEM-1 was susceptible to aztreonam, but not amoxicillin/clavulanate. The control *K. pneumoniae* strain was resistant to both aztreonam and amoxicillin/clavulanate. The other positive control *C. braakii* HDC438 tested positive for KPC-2 and plasmid-mediated AmpC (CMY-79) genes. The MIC values of aztreonam and amoxicillin/clavulanate alone were all greater than or at least 64 mg/L, indicating that these isolates were very resistant to both aztreonam and amoxicillin/clavulanate. The aztreonam/amoxicillin/clavulanate combination has a significant synergistic effect on the control *K. pneumoniae* strain (FICI = 0.0937). This combination reduced the MIC of aztreonam and amoxicillin/clavulanate to values below their clinical breakpoint in 7/9 *K. pneumoniae* and 8/9 *E. coli* [24]. In three isolates, the combination did not significantly reduce the MICs compared to that of monotherapy.

The MPC values of aztreonam and amoxicillin/clavulanate alone or in combination against three *K. pneumoniae* and four *E. coli* isolates are shown in Table 3. The MPC values of aztreonam alone were all > 256 mg/L, whereas the MPCs of amoxicillin/clavulanate alone were ≥128 mg/L. The combination consisting of aztreonam/amoxicillin/clavulanate significantly reduced the MPC values to ≤8 mg/L for these seven isolates. The reduction in MPC was at least 32-fold.

### 3.2. Time–Kill Kinetics

The results of the time–kill kinetics are shown in Figure 1. Amoxicillin/clavulanate as monotherapy at 128/64 and 256/128 mg/L did not reduce the bacterial density of the *K. pneumoniae* LW5 isolate, whereas aztreonam treatment at 32 and 64 mg/L resulted in an initial kill, but rebounded at 4 and 8 h, respectively. For *E. coli* 37, the bacterial burden in the amoxicillin/clavulanate treatment group continued to rise and was not effectively suppressed at 128/64 mg/L, whereas bacteriostasis was observed at 256/128 mg/L. Bacteriostasis was observed at 8h when treated with aztreonam alone at MIC and at two-fold MIC. We did not evaluate the extremely high concentration of amoxicillin/clavulanate or aztreonam required to exert a bactericidal effect against these two isolates. Previous experience indicated that aztreonam MIC against extremely resistant *E. coli* harboring NDM-1 could be as high as 2048 mg/L [33]. 

Treatment with aztreonam/amoxicillin/clavulanate showed that a 2 log_10_-kill was achieved at around 4 h. These isolates did not exhibit a growth rebound at 8 h when treated with aztreonam/amoxicillin/clavulanate, except for *K. pneumoniae* LW5 exposed to amoxicillin/clavulanate plus aztreonam at MIC. Aztreonam/amoxicillin/clavulanate resulted in a bactericidal effect at 8 h even though the concentrations of the combination are much lower than aztreonam or amoxicillin/clavulanate alone.

### 3.3. Pharmacodynamic Analysis of Resistant Mutant Selection in Blood

The clinical dosing regimens of amoxicillin/clavulanate and aztreonam by renal function categories listed in Table 1 were used to evaluate the effects of combination therapy on the PD parameters associated with restricting the emergence of resistant mutants. The doses were expected to achieve ≥ 90% PTA for 60% *f*T_>MIC_ of ≤8 mg/L for aztreonam and 50% *f*T_>MIC_ of ≤ 8 mg/L for amoxicillin/clavulanate in the plasma (Figure 2). A PTA of approximately 80% at 8 mg/L MIC was achieved in patients with creatinine clearance > 50 mL/min using a dosing regimen 1/0.2 g amoxicillin/clavulanate q6h (Figure 2). 

A hypothesis was tested such that the aztreonam/amoxicillin/clavulanate combination will reduce the *f*T_MSW_ and increase *f*T_>MPC_, compared to the corresponding monotherapy. For this hypothesis to work, sufficient PTA (i.e., ≥90%) should be achieved at the isolate combination MICs [34]. These two PD parameters were only determinable in the combination setting because the MIC and MPC in the monotherapy were too high. As shown in Table 4, aztreonam *f*T_>MPC_ was, on average, ≥90%, whereas *f*T_MSW_ was <10% in the plasma. Two *E. coli* isolates, 13 and 48, had higher combination MPC and MIC than other isolates, resulting in lower *f*T_>MPC_ and higher *f*T_MSW_. 

In the combination therapy, the overall *f*T_>MPC_ of amoxicillin/clavulanate was also > 90% except for the two *E. coli* isolates 13 and 48, which had the highest MPC of 8 mg/L. The *f*T_MSW_ were very limited, indicating that the opportunity for the selection of resistant mutants is restricted in the combination therapy. 

### 3.4. Pharmacodynamic Analysis of Resistant Mutant Selection in ELF

A 40% penetration [28] into the ELF was assumed for aztreonam; the two PD indices in the ELF were determined based on a lower drug exposure in the ELF. Apart from *E. coli* isolates 13 and 48, the *f*T_MSW_ was close to 0 for the other isolates (Table 5). For these two isolates, the *f*T_MSW_ ranged from 4 to 10% of the dosing interval. Aztreonam *f*T_>MPC_ was, on average, ≥90% in the ELF. We assumed an ELF penetration rate of 30% for amoxicillin/clavulanate [29,35]. The *f*T_MSW_ was <15% and *f*T_>MPC_ was >60% for all isolates, except *E. coli* 13 and 48 (Table 5). For these two isolates, the *f*T_MSW_ ranged from 10 to >40% of the dosing interval. The combination significantly reduced the likelihood for resistant mutants to emerge in the ELF by restricting the time that the bacteria spent in MSW.

In the computation of PTA for the dosing regimens of aztreonam and amoxicillin/clavulanate in the ELF, we assumed that aztreonam and amoxicillin/clavulanate have very little binding to mucin. Figure 3 shows that aztreonam dosing regimens were expected to achieve ≥ 90% PTA for 60% *f*T_>MIC_ of ≤8 mg/L in the ELF; this breakpoint is the same as that in the plasma. Because amoxicillin and clavulanate have a lower ELF penetration rate, the dosing regimens of amoxicillin/clavulanate were expected to achieve ≥ 90% PTA for 50% *f*T_>MIC_ of ≤4 mg/L in patients with creatinine clearance < 50 mL/min. For the dosing regimen in patients with creatinine clearance > 50 mL/min, a ≥90% PTA for 50% *f*T_>MIC_ was expected to be achieved at MIC of ≤2 mg/L.

## 4. Discussion

The acyl enzyme and β-lactamase inhibitor complex that is formed from nucleophilic attack by clavulanic acid is more stable than β-lactams since clavulanic acid is a suicidal inhibitor, whereas β-lactam antibiotics are hydrolyzed by the enzyme. In contrast, avibactam does not interact via an acyl group, but rather by a carbamyl moiety, resulting in a high potency against various β-lactamases including AmpC, P99, KPC-2, TEM and SHV wherein clavulanic acid potencies against these enzymes are limited [20,36]. The majority of the ESBL genes in our collection of clinical isolates are plasmid-encoded and can be acquired by horizontal gene transfer; these isolates are likely clonally-related, given that they were collected from the same hospital and within a period of 5 years. Most of the plasmid-encoded β-lactamase genes were for SHV, TEM and CTM-M. CMY-42, encoded by the plasmid-mediated AmpC gene [37], may be responsible for the resistance of *E. coli* isolate 9 to aztreonam/amoxicillin/clavulanate combination. The *C. braakii* isolate expressing KPC-2 and CMY-79 enzymes was resistant to the triple antibiotic combination. These two isolates have the ESBL gene wherein clavulanic acid has limited potency; the IC_50_ of clavulanic acid against AmpC and KPC-2 enzymes was >100 nM (Table 6) [20]. As for the *K. pneumoniae* LW1 and LW15 isolates, the identified ESBLs were supposed to be inhibited by clavulanic acid. Resistance to the triple combination in these two isolates could be due to the hyperproduction of these plasmid-encoded β-lactamases [38,39]; this mechanism was not evaluated in our study. In our previous study, the aztreonam/avibactam combination was potent against these four isolates [11], indicating that the deficiency in the OmpF and/or OmpC porins is unlikely the mechanism for their resistance to aztreonam/amoxicillin/clavulanate combination.

For other isolates, the effect of the aztreonam/amoxicillin/clavulanate combination was comparable to the combination of aztreonam/ceftazidime/avibactam in terms of antimicrobial susceptibility [11]. The results suggest that this combination can be used as a limited alternative to aztreonam/ceftazidime/avibactam. Emeraud also measured the in vitro susceptibility of three combinations of aztreonam with ceftazidime/avibactam, ceftazidime/tazobactam, and amoxicillin/clavulanate to multi-drug resistant bacteria, and their results showed that aztreonam/amoxicillin/clavulanate was similar to aztreonam/ceftazidime/avibactam against some bacterial strains and also more cost-effective [40]. The combination of aztreonam and clavulanate may also be promising against MLB-producing Enterobacterales, but we did not evaluate this combination given that this combination or clavulanate alone is not available commercially. We evaluated aztreonam/sulbactam combination against these isolates, but did not observe synergistic activities for this combination (data not shown).

Amoxicillin is the preferred partnering drug for clavulanate over ticarcillin, since clavulanate induces the expression of AmpC and antagonizes the antibacterial activity of ticarcillin in *Pseudomonas aeruginosa* [41]. The aztreonam/amoxicillin/clavulanate combination, however, is not useful against *P. aeruginosa* (data not shown). Chromosomal AmpC hyperproduction in *P. aeruginosa*, being the main mechanism driving β-lactam resistance, is not inhibited by clavulanic acid. In contrast, aztreonam/avibactam could inhibit AmpC and was active against AmpC-producing *P. aeruginosa* [33,42]. The coverage of the aztreonam/amoxicillin/clavulanate combination is limited to *E. coli* and *K. pneumoniae*, depending on the types of β-lactamases expressed in the pathogen, whereas aztreonam/ceftazidime/avibactam may provide broader spectrum coverage of β-lactamases and pathogens.

The current study differentiates from other studies evaluating synergistic activities of aztreonam/amoxicillin/clavulanate combination against NDM- and ESBL-co-producing Enterobacterales in that we examined the notion that effective combination should not only produce synergism, but also restrict the emergence of resistant mutants by reducing the MSW and maximizing *f*T_MSW_. Xu et al. conducted checkerboard microdilution and time–kill assays on 56 strains of carbapenem-resistant Gram-negative pathogens. The combination of amoxicillin/clavulanate and aztreonam was found to have a synergistic effect of up to 97% of *E. coli* [23]. Ract et al. compared the synergistic activity of combinations consisting of aztreonam/amoxicillin/clavulanate or piperacillin/tazobactam in 22 strains of *K. pneumoniae* and 14 strains of *E. coli*. They concluded that the combination of aztreonam/amoxicillin/clavulanate was more effective than that with piperacillin/tazobactam, and this combination could be an attractive unconventional treatment [43]. These existing studies have only evaluated synergistic activities using MIC and compared them with clinical breakpoints. 

To infer the clinical effects of this combination, we conducted simulations using tested and recommended clinical dosing regimens. For aztreonam, we chose the dosing regimens from the REJUVENATE clinical trial rather than the recommended regimens in the package insert; the safety of aztreonam for these dosing regimens has been demonstrated in the trial [26]. For amoxicillin/clavulanate, we chose the clinically recommended dosing regimens. In this in vitro and simulation study, the PD parameters, *f*T_MSW_ and *f*T_>MPC_, were calculated to extrapolate the clinical efficacy of aztreonam/amoxicillin/clavulanate combination and its capacity to restrict the emergence of drug-resistant mutation against bacteria already harboring NDM. In select isolates wherein combination therapy could bring their MICs to their breakpoints, the *f*T_>MPC_ values are greater than 90% and all of the *f*T_MSW_ values are less than 6% in the blood, except for amoxicillin/clavulanate in patients with renal clearance >50 to 150 mL/min. A larger dose of amoxicillin/clavulanate can result in better suppression of further resistance development so that *f*T_>MPC_ is sufficiently large and the *f*T_MSW_ is reduced to <10% of the 24 h duration, especially in the case wherein MPC is 8 mg/L and MIC is 4 mg/L for amoxicillin/clavulanate. Much of the literature suggested that the clinically recommended dose of amoxicillin/clavulanate is too conservative for patients with high renal clearance [27,44]. 

**Table 6 pharmaceutics-15-00251-t006:** IC_50_ (μM) of clavulanic acid and other β-lactamase inhibitors against ESBLs from the literature.

ESBLs	IC_50_ (μM)				Reference
	Clavulanic Acid	Tazobactam	Sulbactam	Avibactam	
P99	>100	1.3	21.1	0.1	Stachyra et al., 2010 [20]
AmpC	>100	4.6	27	0.128	
CTM-M-15	0.012	0.006	0.23	0.005	
TEM-1	0.058	0.032	1.56	0.008	
KPC-2	>100	50	57	0.17	
SHV-4	0.004	0.055	0.26	0.003	
SHV-1	0.06	0.14	3.1		Giakkoupi et al., 1999 [45]
SHV-5	0.02	0.08	1.8		
OXA-24	50	0.5	40		Bou et al., 2000 [22]
TEM-1	0.09	0.04	6.1		Payne et al., 1994 [21]
TEM-2	0.18	0.05	8.7		
TEM-3	0.03	0.01	0.03		
TEM-5	0.03	0.28	1.2		
TEM-6	0.12	0.17	0.45		
TEM-7	0.10	0.18	0.62		
TEM-9	0.29	0.34	0.9		
TEM-10	0.03	0.08	0.34		
SHV-1	0.03	0.14	17		
SHV-2	0.05	0.13	2.8		
SHV-3	0.04	0.1	2.7		
SHV-5	0.01	0.08	0.63		
Enzyme A	0.09	0.07	0.48		
Enzyme B	0.01	0.01	0.12		
Enzyme C	0.33	0.09	10		
Enzyme D	0.04	0.01	0.57		
Enzyme E	0.09	0.11	3.6		
TEM-E1	0.05	0.02	0.64		
TEM-E2	0.09	0.05	1.6		
TEM-E3	0.02	0.06	0.2		
TEM-E4	0.06	0.04	0.79		
CAZ-3	0.13	0.06	2.5		
DJP-1	0.01	0.02	0.21		
TLE-1	0.11	0.05	5.5		
MJ-1	0.09	0.43	40		
PSE-4	0.15	0.1	3.7		
BRO-1	0.02	0.02	0.02		
OXA-1	1.8	1.4	4.7		
OXA-2	1.4	0.01	0.14		
OXA-4	8.4	5.6	16		
OXA-5	3.1	0.25	18		
OXA-6	1.6	1.7	51		
OXA-7	0.36	0.61	40		
PSE-2	0.81	0.94	37		

## 5. Conclusions

This combination could play an important, but limited, niche in the clinical treatment of multidrug-resistant *E. coli*- and *K. pneumonia* carrying MBLs and other β-lactamases. Our study offers a potential scope and limitations as to where the aztreonam/amoxicillin/clavulanate combination may succeed or fail. 

## Figures and Tables

**Figure 1 pharmaceutics-15-00251-f001:**
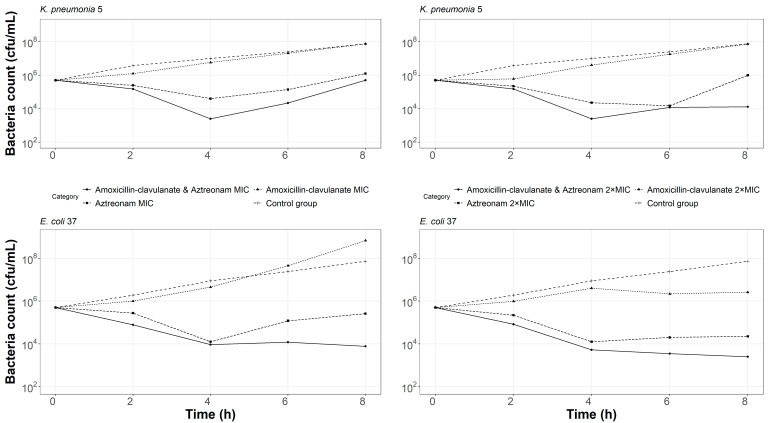
Static-concentration time–kill kinetics of amoxicillin/clavulanate and aztreonam alone and in combination at their respective MIC and two-fold MIC against *K. pneumoniae* 5 and *E. coli* 37.

**Figure 2 pharmaceutics-15-00251-f002:**
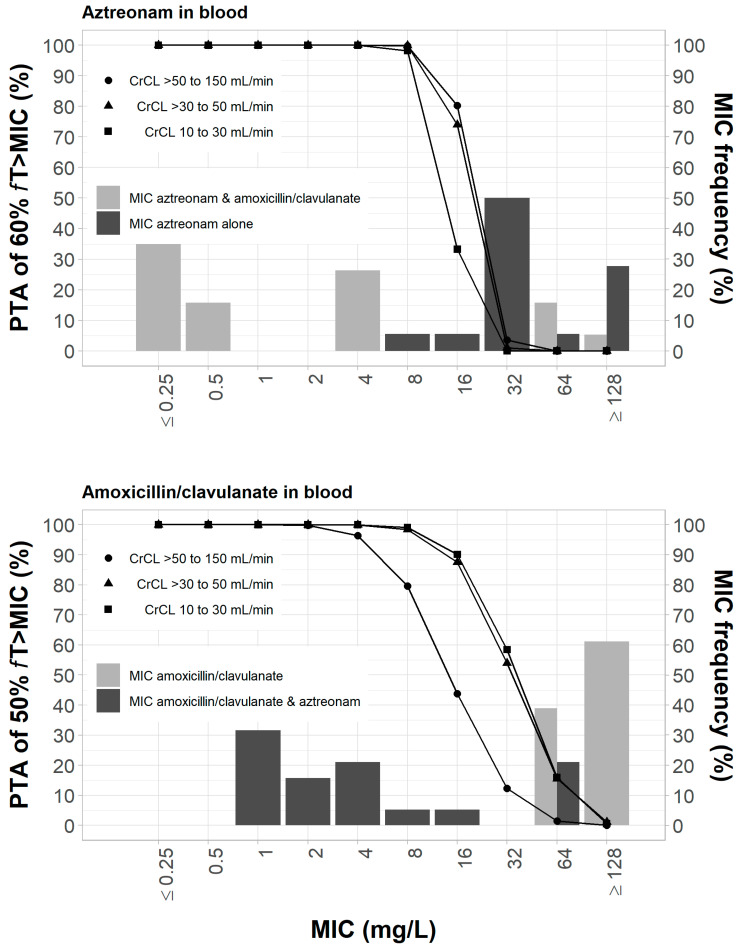
Probability of target attainment of 60% *f*T_>MIC_ and 50% *f*T_>MIC_ for aztreonam and amoxicillin/clavulanate dosing regimens by renal function category, respectively, listed in Table 1. Probability of target attainment values were computed based on steady-state drug concentrations in the blood.

**Figure 3 pharmaceutics-15-00251-f003:**
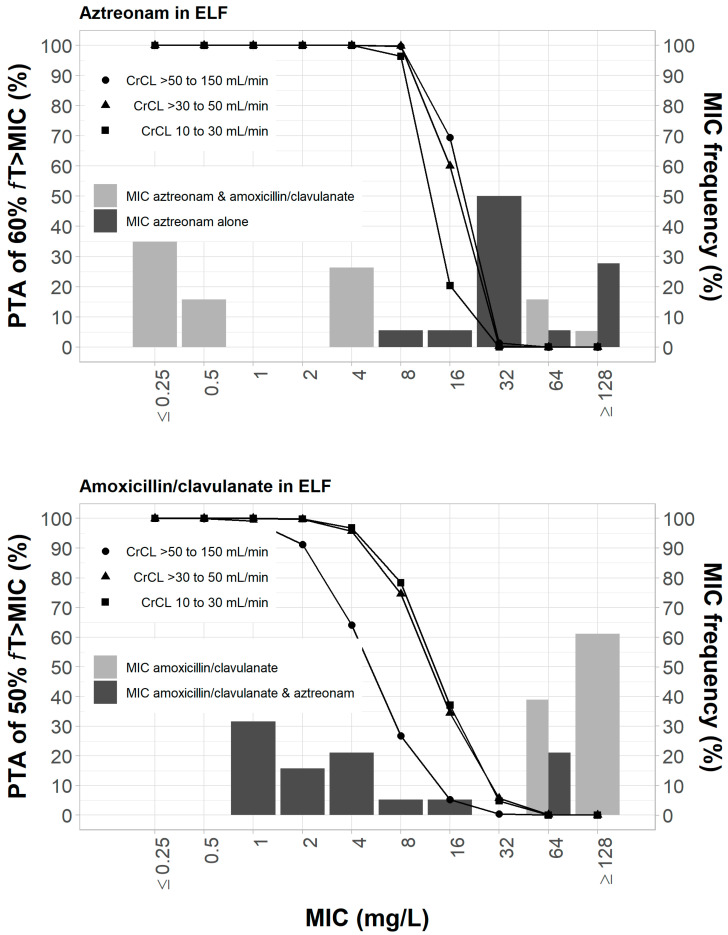
Probability of target attainment of 60% *f*T_>MIC_ and 50% *f*T_>MIC_ for aztreonam and amoxicillin/clavulanate dosing regimens by renal function category, respectively, listed in Table 1. Probability of target attainment values were computed based on steady-state drug concentrations in the epithelial lining fluid and their respective epithelial lining fluid penetration.

**Table 1 pharmaceutics-15-00251-t001:** Dosing regimens of aztreonam and amoxicillin/clavulanate as 3 h infusion used in simulation by creatinine clearance category.

Creatinine Clearance	Aztreonam (High Dose)	Amoxicillin/Clavulanate
>50 to 150 mL/min	2g LD 1.5g MD q6h	1/0.2 g q6h
>30 to 50 mL/min	2g LD 750g MD q6h	1/0.2 g q6h
10 to 30 mL/min	2g LD 500 mg MD q8h	1/0.2 g followed by 500/100 mg q12h

LD, loading dose; MD, maintenance dose.

**Table 2 pharmaceutics-15-00251-t002:** Minimum inhibitory concentrations (MIC) of aztreonam (ATM) alone, amoxicillin/clavulanate (AMC/CA, 2:1 ratio) and aztreonam/amoxicillin/clavulanate (ATM/AMC/CA) against *K. pneumonia* and *E. coli* isolates, as well as β-lactamase genes encoded in each isolate.

Strains	ATM Alone MIC (mg/L)	AMC/CA MIC (mg/L)	ATM/AMC/CA MIC (mg/L)	FICI	β-Lactamase Encoded
Control					
*E. coli* ATCC 25922	0.5	2/1	-	-	
*E. coli* ATCC 35218	0.5	16/2	-	-	
*K. pneumoniae* ATCC 700603	16	32/16	1/1/0.5	0.0937	
*C. braakii* HDC438	>128	>128/64	64/64/32	1	CMY-79, NDM-1, KPC-2
*K. pneumonia*					
LW1	>128	64/32	128/64/32	2.00	TEM-1B, SHV-1, CTX-M-15, NDM-1
LW4	32	64/32	0.25/2/1	0.0391	CTX-M-15, NDM-1, TEM-1B, SHV-1
LW5	32	64/32	0.25/1/0.5	0.0234	NDM-1, TEM-1B, SHV-1
LW8	32	>128/64	0.5/1/0.5	0.0234	CTX-M-15, TEM-1B, SHV-1, NDM-1
LW9	32	>128/64	0.5/2/1	0.0313	SHV-1, CTX-M-15, NDM-1, TEM-1B
LW10	32	>128/64	0.25/4/2	0.0391	CTX-M-15, TEM-1B, NDM-1, SHV-25
LW13	32	64/32	0.25/2/1	0.0391	SHV-1, TEM-1B, NDM-1, CTX-M-15
LW15	>128	>128/64	64/64/32	1.00	SHV-26, TEM-1B, CTX-M-15, NDM-1
LW16	32	>128/64	0.5/1/0.5	0.0234	CTX-M-15, TEM-1B, SHV-1, NDM-1
MIC50	32	>128/64			
MIC90	>128	>128/64			
*E. coli*					
EC9	>128	>128/64	64/64/32	1.00	NDM-1, CMY-42
EC13	>128	>128/64	4/4/2	0.0625	NDM-5, CTX-M-55
EC36	32	>128/64	4/8/4	0.1875	NDM-1, TEM-20, CMY-6, TEM-1C
EC37	32	>128/64	0.25/1/0.5	0.0156	NDM-1, CTX-M-55, TEM-1B
EC38	>128	>128/64	4/16/8	0.1563	NDM-7, OXA-1, CTX-M-55
EC42	16	>128/64	4/4/2	0.2813	TEM-20, NDM-1, CMY-6, TEM-32
EC45	64	64/32	0.25/1/0.5	0.0195	TEM-1B, NDM-13, TEM-141, CTX-M-55, OXA-1
EC48	>128	64/32	4/4/2	0.0938	CTX-M-65, TEM-34, NDM-9, TEM-1B, OXA-1
EC52	8	64/32	0.25/1/0.5	0.0469	TEM-1B, NDM-1
MIC50	32	>128			
MIC90	>128	>128			

MIC, minimum inhibitory concentration; ATM, aztreonam; AMC/CA, amoxicillin/clavulanate; FICI, fractional inhibitory concentration index.

**Table 3 pharmaceutics-15-00251-t003:** Mutant prevention concentrations (MPC) of aztreonam (ATM), amoxicillin/clavulanate (AMC/CA, 2:1 ratio) and aztreonam/amoxicillin/clavulanate (ATM/AMC/CA) against seven *K. pneumonia* and *E. coli* isolates.

	Mutant Prevention Concentration (mg/L)		
	ATM Alone	AMC/CA	ATM/AMC/CA
*K. pneumonia*			
LW5	>256	>256/128	1/1/0.5
LW8	>256	>256/128	1/1/0.5
LW13	>256	128/64	4/4/2
*E. coli*			
EC13	>256	>256/128	8/8/4
EC37	>256	>256/128	2/2/1
EC45	>256	>256/128	1/1/0.5
EC48	>256	>256/128	8/8/4

ATM, aztreonam; AMC, amoxicillin; CA, clavulanate.

**Table 4 pharmaceutics-15-00251-t004:** Pharmacodynamic parameters *f*T_MSW_ and *f*T_>MPC_ computed based on MIC and MPC of aztreonam and amoxicillin/clavulanate as combination therapy in the plasma against *E. coli* and *K. pneumoniae* isolates harboring NDM and serine-β-lactamases.

	Dosing Regimen ^‡^ in CrCL > 50 to 150 mL/min		Dosing Regimen ^‡^ in CrCL > 30 to 50 mL/min		Dosing Regimen ^‡^ in CrCL 10 to 30 mL/min	
Bacteria Isolate	*f*T_MSW_ (Mean ± SE)	*f*T_>MPC_ (Mean ± SE)	*f*T_MSW_ (Mean ± SE)	*f*T_>MPC_ (Mean ± SE)	*f*T_MSW_ (Mean ± SE)	*f*T_>MPC_ (Mean ± SE)
*Aztreonam*						
EC13, EC48	3.2 ± 0.07%	96.3 ± 0.00082%	4.0 ± 0.09%	95.8 ± 0.09%	7.2 ± 0.12%	92.2 ± 0.13%
LW13	0.4 ± 0.023%	99.6 ± 0.023%	0.2 ± 0.015%	99.8 ± 0.02%	0.6 ± 0.03%	99.3 ± 0.03%
EC37	0.12 ± 0.0053%	99.9 ± 0.005%	0.1 ± 0.003%	99.9 ± 0.003%	0.1 ± 0.007%	99.8 ± 0.007%
LW8	0.03 ± 0.0007%	99.9 ± 0.001%	0.04 ± 0.004%	99.9 ± 0.005%	0.03 ± 0.0014%	99.9 ± 0.002%
EC45	0.04 ± 0.0007%	99.9 ± 0.001%	0.02 ± 0.0002%	100 ± 0.0003%	0.05 ± 0.0015%	99.9 ± 0.002%
*Amoxicillin/clavulanate*						
EC13, EC48	14.4 ± 0.16%	75.8 ± 0.29%	2.4 ± 7.7%	96.9 ± 0.11%	1.0 ± 0.06%	98.8 ± 0.07%
LW13	5.8 ± 0.09%	90.0 ± 0.17%	0.5 ± 3.0%	99.4 ± 0.04%	0.1 ± 0.02%	99.8 ± 0.02%
EC37	2.6 ± 0.06%	95.6 ± 0.11%	0.2 ± 1.3%	99.7 ± 0.02%	0.1 ± 0.005%	99.8 ± 0.005%
LW8	0%	98.2 ± 0.07%	0%	99.9 ± 0.007%	0%	99.9 ± 0.002%
EC45	0%	98.2 ± 0.07%	0%	99.9 ± 0.007%	0%	99.9 ± 0.002%

CrCL, creatinine clearance; MIC, minimum inhibitory concentration; MPC, mutant prevention concentration; *f*T_MSW_, fraction of time within 24 h that the drug concentration is within mutant selection window; *f*T_>MPC_, fraction of time within 24 h that the drug concentration is above MPC. ^‡^ See Table 1 for list of dosing regimens by renal function category.

**Table 5 pharmaceutics-15-00251-t005:** Pharmacodynamic parameters *f*T_MSW_ and *f*T_>MPC_ computed based on MIC and MPC of aztreonam and amoxicillin/clavulanate as combination therapy in the epithelial lining fluid against *E. coli* and *K. pneumoniae* isolates harboring NDM and serine-β-lactamases.

	Dosing Regimen ^‡^ in CrCL > 50 to 150 mL/min		Dosing Regimen ^‡^ in CrCL > 30 to 50 mL/min		Dosing Regimen ^‡^ in CrCL 10 to 30 mL/min	
Bacteria Isolate	*f*T_MSW_ (Mean ± SE)	*f*T_>MPC_ (Mean ± SE)	*f*T_MSW_ (Mean ± SE)	*f*T_>MPC_ (Mean ± SE)	*f*T_MSW_ (Mean ± SE)	*f*T_>MPC_ (Mean ± SE)
*Aztreonam*						
EC13, EC48	4.3 ± 0.0008%	94.2 ± 0.1%	5.8 ± 0.11%	93.9 ± 0.12%	9.5 ± 0.13%	89.7 ± 0.15%
LW13	0.6 ± 0.0003%	99.4 ± 0.03%	0.3 ± 0.02%	99.7 ± 0.02%	0.8 ± 0.04%	99.1 ± 0.04%
EC37	0.14 ± 0.0064%	99.8 ± 0.006%	0.1 ± 0.003%	99.8 ± 0.003%	0.2 ± 0.009%	99.8 ± 0.009%
LW8	0.03 ± 0.001%	99.9 ± 0.001%	0.03 ± 0.0002%	99.9 ± 0.0004%	0.04 ± 0.002%	99.9 ± 0.002%
EC45	0.04 ± 0.001%	99.9 ± 0.001%	0.04 ± 0.0003%	99.9 ± 0.0004%	0.05 ± 0.002%	99.9 ± 0.002%
*Amoxicillin/clavulanate*						
EC13, EC48	35.0 ± 0.27%	28.2 ± 0.38%	19.3 ± 0.27%	74.3 ± 0.36%	14.3 ± 0.28%	82.5 ± 0.34%
LW13	22.0 ± 0.2%	63.2 ± 0.36%	4.9 ± 0.12%	93.6 ± 0.17%	2.7 ± 0.11%	96.8 ± 0.13%
EC37	8.9 ± 0.02%	84.5 ± 0.22%	1.4 ± 0.05%	98.0 ± 0.07%	0.6 ± 0.03%	99.1 ± 0.04%
LW8	0%	93.5 ± 0.13%	0%	99.4 ± 0.03%	0%	99.8 ± 0.01%
EC45	0%	93.5 ± 0.13%	0%	99.4 ± 0.03%	0%	99.8 ± 0.01%

CrCL, creatinine clearance; MIC, minimum inhibitory concentration; MPC, mutant prevention concentration; *f*T_MSW_, fraction of time within 24 h that the drug concentration is within mutant selection window; *f*T_>MPC_, fraction of time within 24 h that the drug concentration is above MPC. ^‡^ See Table 1 for list of dosing regimens by renal function category.

## Data Availability

Not applicable.

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
