# Peer review of "Pharmacokinetic/Pharmacodynamic Evaluation of Aztreonam/Amoxicillin/Clavulanate Combination against New Delhi Metallo-β-Lactamase and Serine-β-Lactamase Co-Producing Escherichia coli and Klebsiella pneumoniae"

_pharmaceutics, 2023, doi:10.3390/pharmaceutics15010251_

Round 1

Reviewer 1 Report (Previous Reviewer 2)

General Comments:

The study is interesting, relevant and of general interest to the readers of this journal.

We found this article well written, with a good and clear organization of the contents and adequate methodology for drug kinetic modeling, although some comments will be done focusing on the improvement of this theme.

The cited references are recent and appropriate to the discussion.

Concerning the main factor of novelty, we can refer to the new predictive modeling for possible "multi-drug-resistant" mutations in the stated combination of drugs.

The inclusion of Table 6 was found particularly interesting to easily compare the drug potency and the discussion of the manuscript was improved.

The limitations of the study were added appropriately.

Comments concerning methodology:

#1 The "Mean+/- SE (or SD)" to indicate the uncertainty around the estimated mean is lacking in table 5.

#2 We appreciate the efforts made by the authors to clarify the issues raised about the methodology, however, in our perspective certain aspects should be subject to greater detail in order to have reproducibility.

In the previous version of the manuscript, we found that the concept of reproducibility was misinterpreted by another concept. Reproducibility is contingent only on whether the methods of the computational analysis were transparently and accurately reported and whether that data, code or other materials were used to reproduce the original results. [“Reproducibility and Replicability in Science”. Washington (DC): National Academies Press (US); 2019 May 7.]

We understand the concerns about intellectual propriety before publication, but this issue can be overcome by stating that “the authors can make detailed methodologies available upon request after publication”. Furthermore, is not the publication of the manuscript sufficient to guarantee this intellectual property?  So, a clear summary of the methodology and changes to the protocol used as a basis must be made, especially in this case where the authors refer to Carlier et al. 2013 as a basis. Carlier´s group used "non-linear mixed-effects modeling in two-drugs simulations" and this manuscript used "ordinary differential equation models" in three-drugs combination.

It is noticed that the information provided in 2.4 (lines 145-175) does not provide the changes made to any of the references cited ([25] and [26]). Please clarify what changes needed to be made, when including the combination of the triple combination of drugs (the novelty claimed by the authors!!!).

#3 In the algorithm that aims for the numerical solution of ordinary differential equations, two convergence conditions are necessary [“On the Convergence of Numerical Solutions to Ordinary Differential Equations” Butcher. https://doi.org/10.2307/2004263]: Stability and Consistency. When the complexity of the analysis is increased, numerical convergence is more challenging. The initial values greatly influence the solution of the numerical algorithm.

            #3.1 Was the simulation of the triple drug combination attempted? and stability/ consistency achieved? This simulation is vital since the entire manuscript argues the evaluation of the triple combination.

#3.2 Since the initial values greatly influence the numerical algorithm in parameter computation, we ask the authors to clarify how EC50 initial parameters were chosen (in the differential equation d/dt(eff)_in RxODE script) and what correlation/equivalence was made with the MIC values

Author Response

#1 The "Mean+/- SE (or SD)" to indicate the uncertainty around the estimated mean is lacking in table 5. Response: Table 5 updated to indicate “mean ± SE”. #2 We appreciate the efforts made by the authors to clarify the issues raised about the methodology, however, in our perspective certain aspects should be subject to greater detail in order to have reproducibility. In the previous version of the manuscript, we found that the concept of reproducibility was misinterpreted by another concept. Reproducibility is contingent only on whether the methods of the computational analysis were transparently and accurately reported and whether that data, code or other materials were used to reproduce the original results. [“Reproducibility and Replicability in Science”. Washington (DC): National Academies Press (US); 2019 May 7.] We understand the concerns about intellectual propriety before publication, but this issue can be overcome by stating that “the authors can make detailed methodologies available upon request after publication”. Furthermore, is not the publication of the manuscript sufficient to guarantee this intellectual property? So, a clear summary of the methodology and changes to the protocol used as a basis must be made, especially in this case where the authors refer to Carlier et al. 2013 as a basis. Carlier´s group used "non-linear mixed-effects modeling in two-drugs simulations" and this manuscript used "ordinary differential equation models" in three-drugs combination. It is noticed that the information provided in 2.4 (lines 145-175) does not provide the changes made to any of the references cited ([25] and [26]). Please clarify what changes needed to be made, when including the combination of the triple combination of drugs (the novelty claimed by the authors!!!). Response: We supplied a separate supplemental material that details how the simulations were carried out. Details of ODE for these models are supplied, including a sample code for simulation of concentration-time profiles in RxODE. #3 In the algorithm that aims for the numerical solution of ordinary differential equations, two convergence conditions are necessary [“On the Convergence of Numerical Solutions to Ordinary Differential Equations” Butcher. https://doi.org/10.2307/2004263]: Stability and Consistency. When the complexity of the analysis is increased, numerical convergence is more challenging. The initial values greatly influence the solution of the numerical algorithm. Response: The detailed parameter values, initial state of each compartment in the ODE are available in the supplemental material. In this study, we are not fitting a model to some observed data. But rather we took a model and simulated concentration-time profiles of virtual individuals based on input model parameters and their distributions. #3.1 Was the simulation of the triple drug combination attempted? and stability/ consistency achieved? This simulation is vital since the entire manuscript argues the evaluation of the triple combination. Response: See supplemental material for the details on simulations of the three antibiotics. #3.2 Since the initial values greatly influence the numerical algorithm in parameter computation, we ask the authors to clarify how EC50 initial parameters were chosen (in the differential equation d/dt(eff)_in RxODE script) and what correlation/equivalence was made with the MIC values? Response: We provided an illustration of how the derivation of these pharmacodynamic parameters in relation to the minimum inhibitory concentrations (MIC) is accomplished in the supplemental material. Read on the subsection “Derivation of pharmacodynamic parameters.” These three parameters are: (1) time above the mutant prevention concentration (fT>MPC); and (2) time above minimum inhibitory concentration (fT>MIC) to derive the time at which the free drug concentration is within the mutant selection window (fTMSW). These parameters are computed from each individual simulated concentration-time profile. There are 10,000 sets of these pharmacodynamic parameters, given that this is the size of our simulation.

Reviewer 2 Report (New Reviewer)

1. line number 130-136. Please check the front type and the size

2. Who is the emerging mutant and how did you clarify as such

3. Highlighted text - please remove the highlight

4. Why did you choose the mixture aztreonam/amoxicillin/clavulanate

5. Please increase the resolution of the figires and increase the font size used in numbers of both axis in all figures

Author Response

Reviewer 2

  1. line number 130-136. Please check the front type and the size

Response: The font type and size have been checked and modified

  1. Who is the emerging mutant and how did you clarify as such

Response: Line 315, “emerging mutants” changed to further resistance development.

  1. Highlighted text - please remove the highlight

Response: The highlights have been removed.

  1. Why did you choose the mixture aztreonam/amoxicillin/clavulanate

Response: Line 77-81, we provided an explanation that aztreonam is not hydrolyzed by NDM while clavulanate inhibits many of serine β-lactamases, the combination of aztreonam and amoxicillin/clavulanate can overcome these NDM-producers co-expressing serine β-lactamase. We also evaluated aztreonam with sulbactam and this combination did not result in synergistic effects (lines 281-83).

  1. Please increase the resolution of the figures and increase the font size used in numbers of both axes in all figures

Response: The figures have been updated.

Round 2

Reviewer 1 Report (Previous Reviewer 2)

We kindly thank the authors for their effort in clarifying the requested methodological issues raised in the review of the manuscript.

Reviewer 2 Report (New Reviewer)

The mansucript can be accepted now

This manuscript is a resubmission of an earlier submission. The following is a list of the peer review reports and author responses from that submission.

Round 1

Reviewer 1 Report

The authors present PK simulation assessments of AMOX/CLAV + ATM dosing regimens detailing susceptibility testing, FICI/Synergy, and mutant restriction against NDM producing clinical isolates. While free drug concentrations and PD indices are offered in both plasma and ELF, the mechanistic rationale behind this combination is not explored or sufficiently discussed.

First, the manuscript lacks a concrete definition of the problem beyond the presence of MBLs. Coproduction of ESBLs and horizontal transfer of ESBL genes among these isolates should be stated and described in the introductory sections as well as the discussion sections. The background section would benefit from describing accurately with evidenced citations the historic role of clavulanate and its potential efficacy as a β-lactamase inhibitor. Prior to delving into mutant restriction, there is a significant body of literature describing the efficacy and inhibition by CLAV against β-lactamases that are often concomitantly co-produced or harbored by NDM-producers. 

Second, in leading up to the efficacy of the combination proposed, it is insufficient to merely reference another study. Circumventing the specific β-lactamases harbored by these organisms is not discussed. This is highly different that avibactam based mechanistic considerations since Clavulanate employs first generation irreversible “suicide inhibition” to permanently inactivate β-lactamase via additional chemical reactions in the active site.

-        Please discuss thoroughly the mechanistic rationale and offer a discussion as to where and why this combination was not effective. The efficacy of these mechanism-based inhibitors indeed varies within and between several classes of β-lactamases. This is applicable for SHV-1, TEM-1, and others. There are many comparative studies of TEM- and SHV-derived enzymes, with published IC50s for clavulanate and many enzymes that can perhaps offer insight into where this combination may find success vs failure. Your results will serve to augment this discussion or may otherwise conflict with that rationale altogether. In all cases, the data generated should be placed within proper context of Background, hypothesis, experiment/data, and conclusions. 

The same type of rationale and mechanistic consideration should be extended to the parent compound, amoxicillin. Knowing that Ticarcillin-Clav is available and offers extended range inhibition. Without going into many details, please offer the mechanistic expectation and background to the role of amoxicillin against these isolates. I do not see sufficient discussions to that end beyond a reference of another study where this combination was employed. It could be sufficient to offer evidence that Clav can antagonize the antibacterial effects of ticarcillin as a parent compound which strengthens the argument for Amox as the partnering B-lactam. As a counterargument to the selection of this BL-BLI combination is to highlight the potential for AmpC production by Clav as well as the myriad of studies showing the poor efficacy of amox-clav against ESBL producers.

-        At the offset of this study title, readers will wonder and want to be convinced that amox-clav will perform well against ESBLs. This is the difference between CAZ-AVI + ATM vs Amox-ClAV + ATM. For that reason, a strong section highlighting the efficacy of the combination used in your study needs to be incorporated into this write-up. The purpose is not to convince readers but rather to offer a potential scope and limitations as to where this combination may succeed or fail. With the advent of more available WGS and rapid molecular diagnostics, it is important to define and attempt to describe the perils and pitfalls of this combination. This is especially valid and true against NDM-producers where co-production of ESBLs and β-lactamase variants is expected. With the data available and generated by your work, defining the mechanistic pitfalls and limitations of this combination should be offered thoroughly in an effort and goal to find a specific niche and usage for this makeshift combination.

 Other minor issues:

39 – Would not use the term emerging to describe polymyxins in any context. They are old agents used as last resort for such infections. I expect the authors intended to clarify that polymyxins are often utilized as last resort agents.

Please highlight the concentration of Amoxicillin used in checkerboards and whether this was fixed.  

MSW and other abbreviations are not established. Please use the full instance of abbreviation initially and then consistently abbreviate throughout the manuscript.

Reviewer 2 Report

The study is interesting, relevant and of general interest to the readers of this journal.

We found this article well written, with a good and clear organization of the contents and adequate methodology for drug kinetic modeling.

The cited references are recent and appropriate to the discussion. 

Concerning the main factor of novelty, we can refer to the new predictive modeling for possible "multi-drug-resistant" mutations in the stated combination of drugs. 

Major comments concerning methodology:

#1 - Considering that the manuscript focuses on the inference of kinetic parameters, the data must be clearly stated being "Mean+/- Standard Error (SE)" to indicate the uncertainty around the estimated mean (precision degree of the parameters estimates).

We recommend the following articles on this matter:

               " Standard deviations and standard errors" by Prof. Altman DG      [https://www.ncbi.nlm.nih.gov/pmc/articles/PMC1255808/].

#2 - Regarding the methodology in the inference analysis, we found serious limitations in the reproducibility of the results (by other laboratories). We advise the authors to elaborate further on this point and clarify how the PK and PD simulations were performed. It is important to clarify the changes in the PK-PD models about previous studies (e.g. ref [18]), perhaps putting in a supplement, the models and/or scripts of these changes.

#2 - Since the initial values greatly influence the convergence of the numerical algorithm in PD parameter estimation, we ask the authors to clarify how initial parameters were chosen, namely the EC50. How were the EC50 values ( in the differential equation d/dt(eff) ) estimated? Has any correlation been made with the MIC values? If so, what equivalence was used?

Major comments concerning results:

#3 - Regarding the output of  PD estimates in Tables 4 and 5, some concerns about the precision of the estimates were raised, namely for EC13, EC48 in both Aztreonam and Amoxicillin/clavulanate combination. In these variables, the standard error(SE) surpasses the mean of the estimate for some cases. Sometimes the SE can be "improved" using bootstrap analysis. Interestingly, these findings are particularly evident for {fT MSW}. I wonder why? Do the authors have any explanation for the low precision of these estimates? 

#4 - Regarding the poor precision of  PD estimates in Tables 4 and 5, we suggest including a new column with the "Relative Standard Error _RSE" to indicate the reliability of the estimates' precision. [https://meps.ahrq.gov/survey_comp/ic_precision_guidelines.shtml _Precision Standards Guidelines for Reporting MEPS-IC Descriptive Statistics]